# Risk-Prioritised Versus Universal Medical Nutrition Therapy for Gestational Diabetes: A Retrospective Observational Study

**DOI:** 10.3390/nu17020294

**Published:** 2025-01-15

**Authors:** Roslyn A. Smith, Madeline Boaro, Ka Hi Mak, Vincent Wong

**Affiliations:** 1Diabetes and Endocrine Service, Liverpool Hospital, Sydney, NSW 2170, Australia; 2Department of Dietetics, Liverpool Hospital, Sydney, NSW 2170, Australia; 3School of Clinical Medicine, University of New South Wales, South West Sydney Clinical Campus, Liverpool, NSW 2170, Australia

**Keywords:** gestational diabetes, medical nutrition therapy, risk assessment, health priorities

## Abstract

Background: The optimal application of medical nutrition therapy (MNT) in treating gestational diabetes remains uncertain. MNT involves individualised nutrition assessment and counselling, which is labour-intensive and is not the sole type of intervention offered by clinical dietitians. Objective: To determine whether pregnancy outcomes differed for individuals with gestational diabetes who were offered MNT on a risk-prioritised (RP) versus universal basis. Methods: Observational data from two cohorts of individuals who were offered MNT only if they met the high-risk criteria following general group-based dietary education (RP1, *n* = 369; RP2, *n* = 446) were compared with a baseline cohort who were universally offered at least one MNT consultation (UM, *n* = 649). The RP1 cohort were seen during community-wide COVID-19 restrictions in 2021, while RP2 were seen after restrictions had lifted in 2022. Furthermore, the RP approach primarily utilised telemedicine, while the UM approach was delivered in person. Results: MNT consultations halved under the RP approach (59 vs. 119 sessions per 100 diagnoses for RP2 vs. UM) and saved more than 20 h of dietitian time per 100 diagnoses (95 vs. 73 h for RP2 vs. UM). No significant increases were observed (*p* < 0.05) for any pregnancy outcomes in the RP cohorts compared with the UM cohort, including usage of diabetes medications, maternal weight gain below and above target, early deliveries, induced deliveries, emergency caesarean sections, large- and small-for-gestational-age (SGA) infants, infant macrosomia, neonatal hypoglycaemia and neonatal intensive care admissions. The use of both basal insulin (27% vs. 33%, OR 0.62, 95% CI 0.46 to 0.84) and metformin (6% vs. 10%, OR 0.52, 95% CI 0.31 to 0.88) was lower in the RP1 cohort during pandemic restrictions compared with the UM cohort; however, these differences were not retained in the RP2 cohort. Additionally, there were fewer SGA infants under the RP approach, particularly for the RP2 cohort (6% vs. 11% for RP2 vs. UM, OR 0.55, 95% CI 0.34 to 0.89). Conclusions: Risk-prioritised MNT was a more efficient dietetic service approach to gestational diabetes than the universal MNT model, with comparable pregnancy outcomes. Similar approaches may represent a strategic way to address sustainable health service planning amidst the rising global prevalence of this condition. However, further research is needed to investigate consumer perspectives, wider service impacts and post-partum maternal and child health outcomes.

## 1. Introduction

Gestational diabetes is associated with adverse birth outcomes, such as foetal macrosomia, preterm delivery and neonatal hypoglycaemia, as well as increased healthcare costs [1]. In the United States alone, the direct costs of the condition have been estimated at USD 1.6 billion [2]. Evidence shows that interventions, including dietary advice, blood glucose monitoring and diabetes medications when necessary, can reduce perinatal morbidity and improve maternal quality of life [3]. The global prevalence is estimated at 14% of pregnancies based on WHO diagnostic criteria [4] and is expected to rise, driving the need for more efficient models of care to meet increasing service demands [2,5].

Clinical practice guidelines consistently recommend dietary intervention by qualified dietitians as the first line of treatment [6]. Such intervention is associated with reduced pregnancy complications, although there is insufficient evidence to support one dietary approach over another [7,8]. Furthermore, there is limited evidence regarding the optimal number and format of dietetic interventions [9]. Medical nutrition therapy (MNT)—which involves individualised assessment, diagnosis, intervention, monitoring and evaluation—is widely considered the “gold standard” of clinical dietetic care and is the primary skill set that distinguishes dietitians from other clinicians [10,11,12]. However, the delivery of MNT to individuals is time- and labour-intensive, such that other dietetic approaches are often utilised. Group education, for instance, has been shown to enhance nutrition knowledge, increase service satisfaction and improve dietary behaviours in individuals with gestational diabetes [13,14,15]. However, few studies have compared group education with individual dietitian consultations. One older investigation reported similar gains in knowledge between the two formats [15], while a more recent study found that while insulin use was higher for group attendees, pregnancy outcomes were comparable to those whose initial dietetic intervention was an individual consultation [16].

Evidence to support universal MNT in gestational diabetes over less intense dietetic interventions remains limited. The most widely recognised nutrition practice guideline, published in 2018 by the Academy of Nutrition and Dietetics (A.N.D.) in the U.S. [9], recommends all individuals with gestational diabetes be referred for MNT. This recommendation is based on five studies, none of which compared MNT with a defined alternative dietetic approach [3,17,18,19,20]. Additionally, the guideline advises a minimum of three MNT consultations for all individuals within the space of 4 weeks, and follow-ups every 2–3 weeks thereafter as required. This recommendation is based solely on expert consensus, as the associated systematic review found no evidence to support a specific consultation frequency to improve foetal, neonatal or maternal outcomes [9]. Furthermore, the appropriateness and feasibility of such recommendations outside the U.S. are questionable given global differences in healthcare resources and the fact that the U.S. employs higher glycaemic thresholds for diagnosing gestational diabetes than countries such as Australia [2]. A more recent scoping review of 12 randomised controlled trials reported equivocal clinical outcomes when comparing MNT with general nutrition education for gestational diabetes; however, only two trials included data on birth outcomes [21].

The objective of this study was to determine whether clinical and birth outcomes differed for individuals with gestational diabetes who were offered MNT on a risk-prioritised versus universal basis following general group-based dietary education.

## 2. Materials and Methods

### 2.1. Study Setting

This retrospective observational study was conducted at a metropolitan teaching hospital in a culturally and linguistically diverse area of Sydney, which has one of the highest rates of gestational diabetes in Australia. Since 2018, between 600 and 800 babies have been born each year at this hospital to mothers diagnosed with gestational diabetes, accounting for more than one in six pregnancies. Individuals booked for delivery at the hospital underwent universal screening for gestational diabetes using a one-step 75 g oral glucose tolerance test (OGTT) using 2013 WHO diagnostic criteria [4]. A diagnosis of gestational diabetes was made if the fasting glucose level was >5.0 mmol/L (92 mg/dL), the 1 h glucose level was ≥10.0 mmol/L (180 mg/dL), or the 2 h glucose level was ≥8.5 mmol/L (153 mg/dL).

### 2.2. Dietetic Service Approaches

The dietetic service approaches from 2019 to 2022 are summarised in Table 1. Prior to the COVID-19 pandemic, all people diagnosed with gestational diabetes were offered at least one individualised MNT consultation with a qualified dietitian following general dietary education, which was predominantly delivered through in-person dietitian-led group sessions. Simplified written materials to accommodate low health literacy were also provided. Individuals completed a one-week food and blood glucose diary between the group session and MNT consultation. Additional MNT consultations were offered if individuals were subsequently re-referred for specific concerns, such as suboptimal glycemia. If lifestyle modifications were deemed insufficient to manage glycemia, medication (insulin or metformin) was initiated by the multidisciplinary team.

In 2021, during community-wide COVID-19 restrictions, a risk-prioritised (RP) dietetic service model was introduced. Under this approach, MNT consultations were only offered to individuals meeting pre-defined risk criteria. These criteria included elevated blood glucose levels, the initiation of insulin therapy, poor glycaemic control while on insulin, excessive or inadequate maternal weight gain, foetal growth concerns and suspected inadequate oral intake. Within the gestational diabetes service, telemedicine (via online or telephone consultations) largely replaced in-person visits, and online group education sessions were implemented in the latter half of 2021, replacing in-person sessions.

The RP approach, including telemedicine delivery, continued after the COVID-19 restrictions lifted. Pre-natal obstetric and midwifery services remained largely unchanged in nature, frequency and format throughout the study period. Endocrine and Diabetes Nurse Educator (DNE) services remained unchanged in nature and frequency; however, some of these transitioned from in-person to telemedicine delivery.

### 2.3. Dietetic Workload Measures

Data on the number of MNT consultations, direct patient contact time and total dietitian time per 100 gestational diabetes diagnoses were obtained from electronic dietetic service records and reported for 2019, 2021 and 2022. These years corresponded to the following service approaches: (a) universal MNT in 2019; (b) RP MNT during COVID-19 restrictions in 2021; and (c) RP MNT following the lifting of COVID-19 restrictions in 2022.

### 2.4. Inclusion Criteria for Pregnancy Outcome Comparison

Clinical and birth outcomes of individuals seen under the universal MNT approach (UM) were compared with those seen under the RP approach. Because the RP approach was introduced during community-wide COVID-19 pandemic restrictions, two RP cohorts were included: one during pandemic restrictions (RP1) and the second after the restrictions lifted (RP2). This was to account for any confounding effects of the restriction period. All individuals meeting the inclusion criteria during the respective service periods were included in the comparison.

Inclusion criteria required individuals to have a gestational diabetes diagnosis, a singleton pregnancy and completed birth data at the time of data extraction. Individuals with pre-existing Type 1 or Type 2 diabetes, gestational diabetes diagnosed before 14 weeks of gestation, multiple pregnancies or missing birth data were excluded.

The service periods were as follows: UM—January to December 2019; RP1—February to December 2021; and RP2—February to December 2022. Data extraction was conducted in early 2022 for the UM cohort and in the early months of the subsequent years for RP1 and RP2.

### 2.5. Data Collection

Baseline, clinical and birth data were extracted from a local clinical reporting system, deidentified and collated in cohort spreadsheets on an intention-to-treat whole-service basis. The reporting system continuously collects information from the electronic medical records of all individuals treated for gestational diabetes at the hospital.

The following baseline variables were extracted: maternal age, ethnicity (self-reported), pre-pregnancy weight (self-reported), pre-pregnancy body mass index (BMI), parity, diagnostic oral glucose tolerance test (OGTT) results and previous history of gestational diabetes based on electronic midwifery reports.

Clinical outcomes included the use of diabetes medications (metformin, basal and prandial insulin) and maternal weight gain based on electronic endocrine and midwifery reports from 37–41-week gestation. Maternal weight gain was compared with internationally recognised guidelines [22] to identify those exceeding or falling short of recommended targets based on pre-pregnancy BMI.

Birth outcomes included mode of delivery, preterm birth (<37-week gestation), neonatal hypoglycaemia, admission to neonatal intensive care unit (NICU), large-for-gestational-age (LGA) and small-for-gestational-age (SGA) infants, neonatal macrosomia (>4 kg) and neonatal deaths based on electronic obstetric reports. LGA and SGA were defined as birth weights above the 90th percentile and below the 10th percentile, respectively, for gestational age at delivery, according to Australian growth charts [23] for infants born after 36-week gestation.

### 2.6. Data Analysis

Statistical analyses were performed by using The Jamovi Project (Version 2.3.21) [Computer Software], with significance set to *p* < 0.05. Baseline and outcome data were presented as numbers (percentages) for categorical variables and medians (interquartile ranges) for non-parametric continuous variables. Comparisons between the RP cohorts and the UM cohort were made by using Pearson’s Chi-square test for categorical variables and one-way analysis of variance (ANOVA) for continuous variables, including the Kruskal–Wallis test for non-parametric baseline variables. Categorical outcomes underwent multivariable logistic regression, adjusted for any baseline confounders.

Following primary comparisons for the full cohorts, a subgroup analysis of outcomes was conducted based on three pre-pregnancy BMI categories: (A) 18.0–24.9 kg/m^2^, (B) 25.0–29.9 kg/m^2^ and (C) ≥30 kg/m^2^.

During the study, evidence emerged in the broader literature to support a distinction between insulin-resistant and insulin-deficient subtypes of gestational diabetes, as characterised by differences in BMI [24]. It was too late to incorporate these new subtypes into the risk criteria underpinning the RP approach. However, to contribute to the evidence base for future RP service design, a secondary analysis was performed on a combined sample of all three cohorts, comparing pregnancy outcomes between individuals with a high pre-pregnancy BMI (≥30 kg/m^2^) and those with a lean BMI (18.0–24.9 kg/m^2^).

### 2.7. Ethical Approval and STROBE Checklist

The clinical reporting system from which the data were extracted has ongoing ethical approval from the South Western Sydney Local Health District Human Research Ethics Committee (LNRSSA/14/LPOOL/75; 14/035b LRN). The Strengthening the Reporting of Observational Studies in Epidemiology (STROBE) checklist [25] was used to ensure the transparency and completeness of reporting and is provided in a Appendix A.

## 3. Results

### 3.1. Dietitian Workload

As shown in Table 2, individual dietitian consultations more than halved under the RP MNT approach in both 2021 and 2022. This decrease resulted in savings of 29 and 22 dietitian hours per 100 gestational diabetes diagnoses in 2021 and 2022, respectively, even with the added time required for triaging administration. The mean time per MNT consultation rose by almost 7 min (21%) from 2019 to 2022.

### 3.2. Baseline Data

Data were collected from 649 individuals who received care under the UM approach and 815 individuals under the RP approach, of which 369 were seen during the COVID-19 restriction period in 2021 (RP1) and 446 after the restrictions lifted in 2022 (RP2). These samples represented the majority of gestational diabetes diagnoses within the relevant time periods, though not the entirety, as some individuals had not yet given birth at the time of data extraction, per the inclusion criteria. The following numbers were excluded from each cohort due to gestational age below 14 weeks at diagnosis or multiple pregnancies: 50 for UM; 39 for RP1; 33 for RP2.

Baseline data are summarised in Table 3. There were no significant differences among the three cohorts in maternal age, ethnic background, parity, previous history of gestational diabetes or diagnosis based on 0 h and 2 h OGTT results. However, the RP1 cohort had higher median pre-pregnancy weight (73 kg vs. 68 kg, *p* < 0.001) and BMI (28.0 vs. 26.2 kg/m^2^, *p* < 0.001) compared with the UM cohort, as well as a higher proportion of individuals in the pre-pregnancy BMI ≥ 30.0 kg/m^2^ category (41% vs. 28%). Additionally, a higher percentage of individuals in the RP cohorts were diagnosed on an elevated 1 h OGTT result (63% and 66% for RP1 and RP2, respectively, vs. 56%; *p* = 0.003). No differences were observed in the baseline characteristics of the three cohorts when pre-pregnancy BMI subgroups were analysed separately.

### 3.3. Clinical and Birth Outcomes

Clinical and birth outcomes for the three cohorts are summarised in Table 4. During the COVID-19 restrictions, after adjusting for pre-pregnancy BMI and 1 h OGTT results, the RP1 cohort had lower rates of any insulin use (34% vs. 45%, OR 0.59, 95% CI 0.44–0.78), basal insulin (27% vs. 33%, OR 0.70, 95% CI 0.51–0.96), metformin (17% vs. 21%, OR 0.52, 95% CI 0.31–0.87), induced deliveries (30% vs. 33%, OR 0.68, 95% CI 0.48–0.60) and elective caesarean sections (15% vs. 21%, OR 0.54, 95% CI 0.36–0.82) compared with the UM cohort. The reduced use of insulin and metformin remained statistically significant for the RP1 period after additional adjustment for maternal weight gain, which was also lower in the RP1 period compared with the UM cohort (median weight gain: 9.3 kg vs. 11.4 kg, *p* < 0.001). These outcomes reverted to non-significance post-pandemic restrictions for the RP2 cohort, except for induced deliveries, which remained lower than the UM cohort (27% vs. 33%, OR 0.64, 95% CI 0.47 to 0.88).

There were fewer SGA infants for the RP cohorts compared with the UM cohort, reaching statistical significance during the RP2 period (6% vs. 11%, OR 0.55, 95% CI 0.34–0.89). No other outcomes were significantly different among the cohorts, including prandial insulin use, maternal weight gain above and below target, early delivery, emergency caesarean section delivery, infant birthweight, infant macrosomia, LGA infants, neonatal intensive care admissions, neonatal hypoglycaemia and neonatal deaths.

Subgroup analysis revealed that compared with the UM cohort, total insulin use was significantly lower during the RP1 period for the pre-pregnancy BMI 18–24.9 kg/m^2^ (21% vs. 35%, OR 0.48, 95% CI0.29–0.81) and BMI 25.0–29.9 kg/m^2^ subgroups (36% vs. 53%, OR 0.50, 95% CI 0.30–0.82), while basal insulin use was significantly lower for the BMI 18–24.9 kg/m^2^ subgroup (13% vs. 25%, OR 0.452, 95% CI 0.247–0.826). The incidence of SGA was significantly lower during the RP2 period for the BMI 25.0–29.9 kg/m^2^ subgroup (4% vs. 13%, OR 0.34, 95% CI 0.12–0.91). No significant differences were observed for any other outcomes within the pre-pregnancy BMI subgroups.

### 3.4. High Versus Lean Pre-Pregnancy Body Mass Index

Despite comparable outcomes between the MNT approaches for the pre-pregnancy BMI subgroups, multiple differences in outcomes emerged when all three cohorts were combined as a single sample and individuals with a high BMI (≥30 kg/m^2^) were compared with those with a lean BMI (18.0–24.9 kg/m^2^).

As shown in Table 5, the high-BMI group had significantly greater usage of any insulin (55% vs. 32%, OR 2.53, 95% CI 2.01–3.17), basal insulin (45% vs. 22%, OR 2.74, 95% CI 2.16–3.49), prandial insulin (22% vs. 17%, OR 1.37, 95% CI 1.04–1.81) and metformin (14% vs. 6%, OR 2.69, 95% CI 1.80-4.00). Additionally, the higher BMI group had greater incidence of maternal weight gain above target (50% vs. 19%, OR 3.42, 95% CI 2.65–4.43), induction of labour (34% vs. 27%, OR 1.37, 95% CI 1.07–1.75) and caesarean section birth (24% vs. 17%, OR 1.45, 95% CI 1.12–1.95).

Adverse neonatal outcomes were more prevalent in the high-BMI group, including a fivefold increase in the odds of a large-for-gestational-age (LGA) infant (26% vs. 6%, OR 5.09, 95% CI 3.51–7.39) and an almost fourfold increase in the odds of infant macrosomia (14% vs. 4%, OR 3.83, 95% CI 2.45–6.00). The odds of neonatal hypoglycaemia (21% vs. 16%, OR 1.42, 95% CI 1.07–1.90) and intensive care unit admissions (14% vs. 10%, OR 1.47, 95% CI 1.03–2.08) were also higher.

Conversely, fewer individuals in the high-BMI group gained below the recommended amount of weight during pregnancy (22% vs. 41%, OR 0.38, 95% CI 0.29–0.49) despite the median maternal weight gain being lower than the lean group (9.6 kg vs. 11.8 kg, *p* < 0.001). The high-BMI group also gave birth to fewer SGA infants (4% vs. 12%, OR 0.35, 95% CI 0.22–0.55) and experienced lower rates of spontaneous delivery (27% vs. 43%, OR 0.50, 95% CI 0.39–0.63).

## 4. Discussion

As the global prevalence of chronic diseases and related conditions like gestational diabetes continues to rise, there is an increasing need to explore innovative, value-based service models that optimise resource use. Sustainable health service planning requires a careful analysis of both effectiveness and feasibility within the local context. This study demonstrated that individuals with gestational diabetes who were offered MNT on a risk-prioritised (RP) basis following group-based dietary education achieved similar clinical and birth outcomes to those seen under a more labour-intensive model, which universally offered at least one MNT consultation per pregnancy. Furthermore, the RP approach was delivered primarily via telemedicine, while the UM approach relied on in-person formats. For every 100 gestational diabetes diagnoses, the RP approach saved 29 h of dietitian time during COVID-19 restrictions and almost 22 h post-pandemic restrictions.

### 4.1. Rationale for a Risk-Prioritised MNT Approach

Traditionally, gestational diabetes nutrition practice guidelines have recommended universal MNT [6], which is resource-intensive. However, gestational diabetes is increasingly being recognised as a heterogeneous condition with varying levels of risk [24]. A recent series in *The Lancet* suggested a stratified “precision medicine” approach to gestational diabetes management, whereby more intensive protocols are reserved for those at the highest metabolic risk, while lower-risk individuals could benefit from less intensive interventions [26].

Several gestational diabetes subtypes have recently been defined and may assist with such risk stratification. A recent multi-centre trial found that individuals diagnosed with “early gestational diabetes” (diagnosed before 20-week gestation) experienced higher rates of insulin use and adverse outcomes compared with those with “late gestational diabetes” (diagnosed at 24–28-week gestation) [27]. Moreover, evidence supports a distinction between the insulin-resistant (IR) and insulin-deficient (ID) subtypes. Individuals with IR gestational diabetes—associated with a higher BMI—experience more adverse pregnancy outcomes compared with those with ID gestational diabetes [24]. Studies have found no significant differences in adverse outcomes between treated, lean individuals with ID gestational diabetes and those without gestational diabetes [28,29]. The secondary findings from the present study support this distinction. Individuals with a high pre-pregnancy BMI had fivefold higher odds of an LGA infant and almost fourfold higher odds of infant macrosomia compared with individuals with a lean pre-pregnancy BMI across all three study cohorts. Individuals with a high BMI also exhibited greater usage of diabetes medications and increased rates of induced labour, neonatal hypoglycaemia and admissions to the neonatal intensive care unit.

A further challenge in determining when and how aggressively to intervene in gestational diabetes is that glycaemia at diagnosis shows a linear rather than threshold-dependent association with pregnancy complications [30]. This has led to marked variations in diagnostic criteria worldwide. For instance, countries like Australia, which follow the WHO 2013 guidelines [4], diagnose more individuals at lower risk of complications compared with the U.S., which employs a two-step method and higher diagnostic thresholds [31]. A meta-analysis estimated that adopting the WHO 2013 criteria in the U.S. would increase gestational diabetes prevalence from 4.9% to 11.5% [32]. However, studies have failed to show improvements in whole-population pregnancy outcomes when a larger number of individuals are diagnosed and treated for gestational diabetes [2]. A 2020 study of over 120,000 Australian pregnancies, for example, found minimal differences in maternal and infant outcomes before and after a tightening of diagnostic criteria, despite a significant increase in gestational diabetes diagnoses [33].

Given this, it is debatable whether MNT is the most appropriate dietetic intervention for lower-risk individuals who meet the WHO 2013 criteria but would not be diagnosed under less stringent thresholds used in other countries. Notably, the diagnostic thresholds used in the five studies supporting the recommendation for universal MNT in the A.N.D. guideline [9] were above the WHO 2013 criteria [3,17,18,19,20]. Two of the studies included even higher risk individuals, with pre-existing Type 2 diabetes comprising more than half the study sample for one [19], while the other was limited to individuals with a history of insulin-requiring gestational diabetes [20].

### 4.2. Pros and Cons of Risk-Prioritised Versus Universal MNT

In this study, the RP approach prioritised MNT based on ongoing triage rather than diagnostic criteria. However, no deterioration in pregnancy outcomes at a service level was observed, despite theoretical concerns—such as the potential to miss higher-risk individuals due to limitations in triage criteria, variability in clinicians’ implementation of the triage process and challenges associated with delivering care via telemedicine. This is in line with other Australian studies which have reported either mixed or statistically non-significant associations between pregnancy outcomes and number of MNT consultations. A 2023 study found higher rates of birth complications, yet lower rates of neonatal intensive care admissions for babies born to individuals who received three compared with zero to two MNT consultations [34]. Three other studies did not find statistically significant associations with insulin rates for services offering at least three MNT consultations per pregnancy as compared with ad hoc care [35,36,37].

Nevertheless, a reduction in MNT consultations may still warrant concern at an individual level. For instance, nutritional deficiencies not identified by the RP triage criteria might have been detected by the UM approach. Additionally, some individuals may have had nutrition-related questions they felt more comfortable discussing in an individual consultation rather than a group session. Multiple consultations also allow for the reinforcing of advice and the provision of progressive information, such as dietary considerations post-partum to reduce the risk of Type 2 diabetes.

Qualitative research outside gestational diabetes suggests that clients are more likely to respond to counselling and therapeutic nutritional approaches to MNT than to a standard educational and informative approach [38]. With sufficient time, MNT can also incorporate advanced coaching techniques, such as motivational interviewing, which can improve lifestyle behaviours in both gestational and Type 2 diabetes [39,40].

However, this highlights a major concern with a universal MNT approach which may limit its effectiveness: the quality of therapy may be diminished by the pressure to provide such an intensive service to a large number of individuals (“production-line” approach). In the current study, only one MNT consultation, with an average duration of approximately 30 min, could be offered per individual under the UM approach, secondary to resourcing constraints. Moreover, the subsequent monitoring and evaluation component of MNT was primarily undertaken by non-dietetic clinicians. By comparison, the A.N.D. guideline recommends 60–90 min for an initial MNT session followed by at least two further 15–45 min follow-up MNT sessions [9].

Under the RP approach, there was a modest seven-minute increase in the average time spent per MNT consultation. However, it was beyond the scope of the study to determine (1) whether individuals at the highest risk received more comprehensive MNT, including multiple sessions, and (2) whether additional universal MNT consultations might have yielded better outcomes than the RP approach. Based on the previously mentioned studies [34,35,36,37], there is limited evidence to justify the substantial increase in resources that would be required to provide additional universal MNT.

Furthermore, providing MNT to a large number of individuals with gestational diabetes may compromise the quality and accessibility of care for other individuals requiring more intensive therapy, such as those with Type 1 and Type 2 diabetes within a broader diabetes service. To the authors’ knowledge, no detailed studies have yet focused on the impact of rising gestational diabetes workloads on services available for other types of diabetes and clinical conditions.

Concerns also exist over potential adverse effects of intensive interventions, such as MNT, for some individuals. Such effects may include undue anxiety, dietary overzealousness and poorer foetal outcomes, such as growth restriction in susceptible individuals [2]. Qualitative research has linked multiple mental health concerns with gestational diabetes management, including the triggering of disordered eating stemming from the “medicalisation” of food [41]. Furthermore, a recent study found an increased rate of SGA infants born to individuals who were treated for gestational diabetes despite being only marginally above the diagnostic threshold in early gestation [27]. This concern may be supported by the present study, which found a lower SGA rate under the RP approach compared with the more intense UM model, which was the most pronounced for the BMI 25.0–29.9 kg/m^2^ subgroup in the RP2 cohort.

### 4.3. Associations During the COVID-19 Pandemic Restriction Period

An interesting finding was the reduction in the use of diabetes medications, maternal weight gain, induced deliveries and elective caesarean deliveries, alongside a higher median pre-pregnancy BMI during the COVID-19 restrictions. These findings were unlikely attributable to the MNT approach, as they largely reverted to non-significance for the RP2 cohort after restrictions had lifted. There are several potential explanations.

Firstly, there may have been reluctance among both healthcare consumers and clinicians to schedule or attend appointments perceived as “non-essential” during the pandemic restrictions (such as an oral glucose tolerance test or insulin commencement), coupled with concerns about exposure to COVID-19. This is consistent with research showing reductions in antenatal care usage generally in Australia during the pandemic, including a 15% reduction in face-to-face services [42].

Secondly, there is the question of falsification of blood glucose levels, which is more challenging to detect via telemedicine compared with in-person consultations, where memory functions can be accessed from monitoring devices. Such falsification has been described in children with diabetes [43] but not in gestational diabetes to the authors’ knowledge. If this had been a significant issue, lower rates of medication use should have persisted under the RP2 approach, as this was also delivered primarily via telemedicine. However, this phenomenon was not observed.

Thirdly, positive lifestyle changes may have resulted from the pandemic restrictions themselves, such as increased time to prepare meals, exercise and maintain a stable routine during “lock-down” periods where non-essential travel, activities (such as eating out) and household visitors were limited. Other Australian research has found improved perinatal outcomes during the pandemic restrictions for pregnant individuals, notably a reduction in preterm births; however, authors suggested this may have been linked with reduced exposure to infections and environmental pollutants rather than self-care behaviours [44].

### 4.4. Study Strengths and Limitations

The strengths of this research study included the substantial sizes of the study cohorts, the “real-world” clinical setting of a staffing-constrained service and the culturally and linguistically diverse population. Additionally, a comprehensive range of clinical and birth outcomes were collected on an intention-to-treat basis, and the RP2 cohort was included, which revealed some confounding effects of the COVID-19 restriction period on the study outcomes.

The study’s limitations included the lack of data on actual numbers of “higher-risk” referrals, reasons for referral, MNT attendance rates and quality of MNT provided for different risk profiles, secondary to the intention-to-treat design. Based on the numbers of MNT consultations per 100 diagnoses, the proportion of “higher risk” individuals was likely below 60%. However, some individuals may have attended multiple MNT consultations, while others, despite referral, may not have attended any. Even if referral and attendance data had been available, a spurious positive association between the number of MNT consultations and adverse outcomes would have been likely due to the confounding effect of higher-risk individuals being flagged for MNT. Nonetheless, further insights—such as potential challenges in attending in-person services compared with telemedicine—might have emerged.

Additionally, the study did not evaluate participants’ nutrition knowledge, dietary behaviours or satisfaction with care, nor did it examine the impact of service changes on other clinicians, such as Diabetes Nurse Educators, who were involved in triaging and often address basic dietary concerns within their consultations.

Furthermore, the long-term effects of MNT on maternal and child health outcomes post-partum will not be fully captured by focusing solely on pregnancy outcomes. Indeed, there is increasing recognition of the importance of integrating gestational diabetes prevention, pregnancy management and post-partum care into a “life course” approach to the underlying metabolic disorder, in order to address its multi-generational impacts [26].

## 5. Conclusions

This study found that offering MNT on a risk-prioritised basis yielded comparable clinical and birth outcomes to a universal MNT approach. The RP approach reduced the number of individual dietitian consultations and represented a more efficient use of dietetic hours, without compromising pregnancy outcomes. Additionally, the study demonstrated sustainability of a new service model, including telemedicine delivery, beyond the restrictions imposed during the COVID-19 pandemic. Further research is needed to determine whether these findings can be generalised to other practice locations.

Moving forward, future research into RP approaches should explore multi-dimensional service impacts, alongside both pregnancy and post-partum outcomes. In particular, studies should investigate the optimal use of MNT alongside less intensive dietary interventions for subtypes of gestational diabetes with differing risk profiles. Future risk-prioritised models could be designed to “predict” higher-risk individuals and provide preventative MNT for gestational diabetes, in contrast to the triaging approach used in this study, which primarily identified individuals who had already developed suboptimal clinical parameters, such as elevated glycemia. The growing recognition of gestational diabetes subtypes, alongside the secondary BMI findings from this study, suggests three key predictors: (1) early gestational diabetes diagnosis, (2) higher diagnostic oral glucose tolerance test (OGTT) results and (3) high pre-pregnancy BMI.

Furthermore, incorporating data on participants’ nutritional knowledge, dietary behaviours and satisfaction with care would provide a more comprehensive evaluation of the effectiveness of MNT service models, alongside analyses that account for confounding factors, such as lifestyle changes and variations in healthcare access.

Finally, assessing the long-term impacts on maternal and child health outcomes post-partum would provide stronger evidence regarding the sustainability of MNT approaches.

## Figures and Tables

**Table 1 nutrients-17-00294-t001:** Universal versus risk-prioritised approaches to medical nutrition therapy in gestational diabetes.

	Universal Medical Nutrition Therapy (UM)	Risk-Prioritised Medical Nutrition Therapy	
	January to December 2019	With Pandemic ^a^ Restrictions (RP1) February to November 2021	Without Pandemic ^a^ Restrictions (RP2) February to November 2022
Written nutrition information package	✓	✓	✓
**↓** General dietitian education (group sessions for English, Arabic and Vietnamese languages and individual sessions for other languages)	In person	In person in February to June and online in July to November	Online for English In person for other languages
**↓** One-week food and glucose diary	✓	✓	✓
**↓** Triaging clinician	Not applicable	Dietitian	Diabetes Nurse Educator (DNE)
**↓** Individual Medical Nutrition Therapy (MNT) consultation	Universally offered	If risk criteria ^b^ met	If risk criteria ^b^ met
**↓** Ongoing monitoring by DNE, endocrine physician or midwife and re-referral for MNT if risk criteria ^b^ met at any stage	✓	✓	✓

^a^ COVID-19 pandemic. ^b^ Risk criteria: elevated blood glucose levels (>20% of readings at any daytime testing point; fasting > 5.2 mmol/L; 2 h post-prandial > 6.9 mmol/L), insulin commencement, suboptimal blood glucose levels on insulin or metformin, suboptimal or excessive rate of weight gain (<0.5 kg or >2.5 kg over the last 4 weeks if >20-week gestation and excluding first week after initial education), foetal growth concerns (estimated foetal weight < 5% or >10% for age and or foetal abdominal circumference more than 2 weeks ahead of other parameters on ultrasound) and suspected inadequate oral intake (frequent problematic hunger, major carbohydrate avoidance reported by patient and gastrointestinal symptoms significantly impacting intake). ✓ means “This was provided”; **↓** shows a sequential flowchart progression.

**Table 2 nutrients-17-00294-t002:** Consult numbers and dietitian time per 100 gestational diabetes diagnoses under a universal versus risk-prioritised medical nutrition therapy (MNT) approach.

	Universal MNT (UM) 2019	Risk-Prioritised MNT			
		With Pandemic ^a^ Restrictions 2021		Without Pandemic ^a^ Restrictions 2022	
			Difference vs. UM		Difference vs. UM
Number of MNT consultations ^b^	118.78	56.50	−62.28 (−52.43%)	59.19	−59.59 (−50.17%)
Direct hours with patients Minutes per consult (mean)	64.07 32.36	34.08 36.19	−29.99 (−46.81%) +3.83 (+11.80%)	38.65 39.18	−25.42 (−39.68%) +6.82 (+21.08%)
Total dietitian hours ^c^	95.38	66.38	−29.00 (−30.40%)	73.49	−21.89 (−22.95%)

^a^ COVID-19 pandemic. ^b^ Greater than 20 min duration. ^c^ Included triage and documentation.

**Table 3 nutrients-17-00294-t003:** Baseline characteristics of individuals with gestational diabetes who were offered medical nutrition therapy (MNT) on a risk-prioritised versus universal basis.

	Universal MNT (UM)	Risk-Prioritised MNT (RP1) ^a^	(RP2) ^b^	*p*-Value ^c^
Number of diagnoses	649	369	446	
Age (years)	31 (28–35)	32 (28–36)	31 (28–35)	0.258
Pre-pregnancy weight (kg) ^d^	68 (58–81)	73 (61–88)	69 (59–84)	0.001 *
Pre-pregnancy BMI ^e^ (kg/m^2^)	26.2 (23.0–30.5)	28.0 (23.8–33.7)	27.2 (23.4–31.9)	<0.001 *
Pre-pregnancy BMI ^e^ subgroups				0.010 *
- <18.0 kg/m^2^	13 (2.00%)	5 (1.36%)	8 (1.80%)	
- 18.0–24.9 kg/m^2^	246 (37.90%)	119 (32.25%)	166 (37.30%)	
- 25.0–29.9 kg/m^2^	206 (31.74%)	95 (25.75%)	124 (27.87%)	
- ≥30.0 kg/m^2^	184 (28.35%)	150 (40.65%)	481 (32.88%)	
Nulliparous	164 (25.27%)	92 (24.93%)	107 (24.10%)	0.906
Previous gestational diabetes	154 (23.73%)	87 (23.64%)	125 (28.15%)	0.195
Oral glucose tolerance test				
- Fasting > 4.9 mmol/L	289 (44.74%)	161 (43.87%)	188 (42.34%)	0.736
- 1 h >9.9 mmol/L	351(55.98%)	233 (63.17%)	284 (65.89%)	0.003 **
- 2 h >8.0 mmol/L	265 (42.20%)	157 (44.48%)	188 (43.32%)	0.783
Ethnicity ^d^				0.256
- South Asia	170 (26.28%)	89 (24.32%)	115 (25.84%)	
- Middle East	173 (26.74%)	84 (22.95%)	106 (23.82%)	
- Southeast Asia	120 (18.55%)	67 (18.31%)	71 (15.96%)	
- Anglo-European	104 (16.07%)	66 (18.03%)	68 (15.28%)	
- Pacific Islands	47 (7.26%)	37 (10.11%)	47 (10.56%)	
- Other	33 (5.10%)	23 (6.29%)	38 (8.53%)	

^a^ With COVID-19 restrictions; ^b^ without COVID-19 restrictions; ^c^ Pearson’s Chi-square 3-cohort combined analyses; ^d^ self-reported; ^e^ body mass index. Data reported as medians (interquartile ranges) for continuous data (none were found to be parametric) or numbers (%) for categorical data. * *p* < 0.001 for RP1 versus UM, non-significant for RP2 versus UM using Pearson’s Chi-square 2 × 2 analyses. ** *p* < 0.05 for RP1 versus UM and *p* < 0.001 for RP2 versus UM using Pearson’s Chi-square 2 × 2 analyses.

**Table 4 nutrients-17-00294-t004:** Clinical and birth outcomes for individuals with gestational diabetes who were offered medical nutrition therapy (MNT) on a universal versus risk-prioritised basis.

	Universal MNT (UM)	Risk-Prioritised MNT		Odds Ratio (95% CI) ^c^	
		(RP 1) ^a^	(RP2) ^b^	RP1 vs. UM	RP2 vs. UM
Number of diagnoses	649	369	446		
**Clinical Outcomes**					
Any insulin	298 (45.92%)	127 (34.42%)	222 (49.89%)	0.536 (0.405–0.709) ***	1.072 (0.833–1.379)
				0.585 (0.439–0.780) ***^d^	1.197 (0.918–1.562) ^d^
- BMI ^e^ subgroup 18.0–24.9 kg/m^2^	86/246 (34.96%)	25/119 (21.01%)	57/166 (34.34%)	0.478 (0.285–0.805) **	0.913 (0.600–1.394)
- BMI ^e^ subgroup 25.0–29.9 kg/m^2^	109/206 (52.91%)	34/95 (35.79%)	72/124 (58.06%)	0.495 (0.300–0.818) **	1.230 (0.783–1.931)
- BMI ^e^ subgroup ≥30.0 kg/m^2^	100/184 (54.35%)	68/150 (45.33%)	92/147 (62.59%)	0.689 (0.447–1.064)	1.015 (0.983–1.047)
Basal insulin	216 (33.28%)	98 (26.56%)	170 (38.20%)	0.621 (0.460–0.837) **	1.129 (0.868–1.468)
				0.701 (0.512–0.958) *^d^	1.315 (0.991–1.744) ^d^
- BMI ^e^ subgroup 18.0–24.9 kg/m^2^	62/246 (25.20%)	16/103 (13.45%)	40/166 (24.10%)	0.452 (0.247–0.826) *	0.897 (0.565–1.424)
- BMI ^e^ subgroup 25.0–29.9 kg/m^2^	77/206 (37.38%)	28/95 (29.47%)	54/124 (43.55%)	0.702 (0.416–1.187)	1.297 (0.823–2.045)
- BMI ^e^ subgroup ≥30.0 kg/m^2^	76/184 (41.3%)	54/150 (36.00%)	76/147 (51.70%)	0.791 (0.507–1.234)	1.505 (0.972–2.332)
Prandial insulin	139 (21.42%)	62 (16.80%)	93 (20.90%)	0.722 (0.511–1.020)	0.922 (0.678–1.254)
Metformin	64 (9.86%)	22 (5.98%)	55 (12.36%)	0.519 (0.306–0.879) *	1.221 (0.815–1.829)
				0.521 (0.311–0.873) *^,d^	1.148 (0.765–1.720) ^d^
Maternal weight gain (kg)	11.40 (7.95–15.00)	9.30 (6.00–14.43)	11.00 (7.00–14.63)	(*p* < 0.001) ***	(*p* = 0.736)
- Below target for BMI ^e^	177 (27.48%)	124 (34.25%)	117 (28.96%)	1.308 (0.951–1.800)	1.183 (0.861–1.625)
- Above target for BMI ^e^	234 (36.34%)	114 (31.49%)	157 (38.86%)	0.819 (0.595–1.126)	1.135 (0.842–1.531)
**Birth Outcomes**					
Early delivery (<37-week gestation)	43 (6.80%)	16 (5.02%)	40 (8.99%)	0.744 (0.409–1.352)	1.349 (0.852–2.134)
Mode of delivery					
- Induction	211 (33.44%)	94 (29.56%)	119 (26.74%)	0.680 (0.482–0.595) *	0.642 (0.469–0.878) **
- Caesarean section—elective	132 (20.92%)	47 (14.78%)	105 (23.60%)	0.544 (0.358–0.824) **	0.927 (0.663–1.298)
- Caesarean section—emergency	71 (11.25%)	54 (16.99%)	49 (11.01%)	1.220 (0.794–1.876)	0.824 (0.539–1.259)
Neonatal birthweight	3306 (3000–3608)	3341(2975–3689)	3283 (2985–3609)	(*p* = 0.424)	(*p* = 0.479)
- Macrosomia (>4 kg)	44 (7.00%)	31 (9.78%)	38 (8.54%)	1.221 (0.723–2.061)	1.209 (0.748–1.953)
- Large for gestational age	84 (13.82%)	52 (16.72%)	66 (16.06%)	1.073 (0.712–1.614)	1.126 (0.774–1.638)
- Small for gestational age	66 (10.86%)	23 (7.40%)	25 (6.08%)	0.753 (0.456–1.246)	0.548 (0.336–0.894) *
- BMI ^e^ subgroup 18.0–24.9 kg/m^2^	31/235 (13.19%)	8/102 (7.84%)	14/154 (9.09%)	0.560 (0.247–1.270)	0.707 (0.360–1.385)
- BMI ^e^ subgroup 25.0–29.9 kg/m^2^	24/191 (12.57%)	8/79 (10.13%)	5/114 (4.39%)	0.828 (0.353–1.942)	0.336 (0.124–0.910) *
- BMI ^e^ subgroup ≥30.0 kg/m^2^	10/171 (5.85%)	5/125 (4.00%)	5/135 (3.70%)	0.694 (0.230–2.090)	0.633 (0.211–1.905)
Neonatal Outcomes					
- Intensive care admission	68 (10.81%)	26 (8.18%)	56 (12.61%)	0.697 (0.425–1.144)	1.148 (0.778–1.695)
- Hypoglycaemia	122 (19.4%)	55 (17.41%)	75 (16.93%)	0.846 (0.592–1.211)	0.801 (0.579–1.106)
- Death	4 (0.63%)	1 (0.31%)	1 (0.22%)	0.450 (0.049–4.104)	0.320 (0.035–2.895)

Reported as medians (interquartile ranges) for continuous data (none were found to be parametric) OR numbers (%) for categorical data. * *p* < 0.05, ** *p* < 0.01 and *** *p* < 0.001. ^a^ With COVID-19 restrictions; ^b^ without COVID-19 restrictions. ^c^ Corrected for baseline pre-pregnancy body mass index and 1 h OGTT; ^d^ additionally corrected for maternal weight gain. ^e^ Body mass index. The bold was to highlight the two main categories of outcomes.

**Table 5 nutrients-17-00294-t005:** Clinical and birth outcomes for individuals with gestational diabetes and lean versus high pre-pregnancy body mass index (BMI).

	Pre-Pregnancy BMI		Odds Ratio
	Lean (18.0–24.9 kg/m^2^; *n* = 672)	High (≥30 kg/m^2^; *n* = 620)	(95% CI) High vs. Lean
**Clinical Outcomes**			
Insulin			
- Any	219 (32.59%)	341 (55.00%)	2.53 (2.01–3.17) **
- Basal	152 (22.62%)	276 (44.52%)	2.74 (2.16–3.49) **
- Prandial	116 (17.26%)	138 (22.26%)	1.37 (1.04–1.81) *
Metformin	38 (5.65%)	86 (13.87%)	2.69 (1.80–4.00) **
Maternal weight gain (kg)	11.8 (8.8–15.0)	9.6 (5.0–14.0)	(*p* < 0.001) **
- Above target	124 (19.28%)	302 (49.92%)	3.42 (2.65–4.43) **
- Below target	265 (41.21%)	137 (22.64%)	0.38 (0.29–0.49) **
- On target	254 (39.5%)	166 (27.44%)	0.79 (0.63–1.0)
**Birth Outcomes**			
Mode of delivery			
- Induction	178 (27.38%)	201 (34.01%)	1.37 (1.07–1.75) *
- Elective caesarean section	112 (17.23%)	139 (23.52%)	1.45 (1.12–1.95) *
- Emergency caesarean section	79 (12.15%)	86 (14.55)%	1.23 (0.89–1.71)
- Spontaneous	277 (42.62%)	160 (27.07%)	0.50 (0.39–0.63) **
Early delivery (<37-week gestation)	38 (5.85%)	49 (8.29%)	1.46 (0.94–2.26)
Small-for-gestational-age babies	74 (11.94%)	25 (4.48%)	0.35 (0.22–0.55) **
Large-for-gestational-age babies	40 (6.45%)	145 (25.99%)	5.09 (3.51–7.39) **
>4 kg babies	27 (4.17%)	84 (14.29%) *	3.83 (2.45–6.00) **
Neonatal hypoglycaemia	103 (15.9%)	124 (21.20%)	1.42 (1.07–1.90) *
NICU	63 (9.72%)	80 (13.63%)	1.47 (1.03–2.08) *
Neonatal death	3 (0.46%)	4 (0.68%)	1.47 (0.33–6.59)

Combined dataset of 2019–2022. Reported as *n* (%) for categorical variables and medians (interquartile ranges) for continuous variables. * *p* < 0.05 and ** *p* < 0.001. The bold was to highlight the two main categories of outcomes.

## Data Availability

The data that support the findings of this study are available from the corresponding author upon reasonable request.

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
