# Peer review of "Risk-Prioritised Versus Universal Medical Nutrition Therapy for Gestational Diabetes: A Retrospective Observational Study"

_nutrients, 2025, doi:10.3390/nu17020294_

Round 1
Reviewer 1 Report
Comments and Suggestions for Authors
Dear Authors,
First of all, I would like to express my sincere gratitude for the opportunity to contribute my feedback to the evaluation of your manuscript. I found the topic addressed to be extremely interesting and relevant to our field. The research presents several useful and promising insights that could lead to significant advancements in our sector. However, after a thorough reading, I believe there are some aspects, mainly methodological, that need to be improved and clarified in order to fully highlight the value of the proposed work. Below, I outline the main areas that could benefit from further exploration and revision.
Editing:
There is no legend for the acronyms used in the summary tables, and the use of acronyms is inconsistent.
Title:
I suggest making it clearer, possibly by shortening it but making it more specific in terms of the study
conducted, e.g., retrospective observational study. Additionally, as in the rest of the text, I recommend removing "mellitus" from the definition of gestational diabetes, as it is now outdated in the scientific community. I advise the authors to review the titles of the most recent prestigious reference guidelines.
Abstract:
There is a lack of a real interpretation in the conclusions regarding the clinical practice implications of the phenomenon studied, with potential suggestions and specific proposals. The results should include (given the type of study conducted) more quantitative data rather than qualitative descriptions. I suggest revising the text (lines 19 to 26) based on the previous comments.
Keywords:
I recommend four or five keywords that are more focused on the main topics of the study; the current ones are too sensitive.
Objectives:
I believe they should conclude the introduction (lines 79-88), while the last part (89-99) should be part of the methods section.
Methods:
This is the most controversial aspect and certainly requires more attention, namely the lack of a structured reporting method, such as the STROBE checklist, which could have been used to ensure the scientific validity and transparency of the study. Including this checklist as a supplementary file and citing it in the text would be crucial to improving the quality and reproducibility of the study, making it more transparent and replicable.
In this regard (lines 100-175), following the checklist mentioned above, I suggest dividing Chapter 2 into subchapters to make it clearer and more fluid.
Results:
Clear and well-reported.
Discussion:
From lines 320 to 350, it is unclear whether the citations at the bottom justify the entire section. I suggest expanding or at least clarifying the references in the text. The same applies, but even more so, from lines 351 to 388, which completely lack supporting references for comparison. This is the most lacking aspect in this section, which necessarily needs to be compared with other studies or different care settings. From lines 403 to 410, they could support the conclusions.
Limitations:
Every research study has limitations that should be explored and discussed clearly and comprehensively. The manuscript does not have a section explicitly dedicated to limitations, and this warrants a dedicated section.
References:
Should be expanded in the critical points mentioned and updated for those older than twenty years, especially in terms of clinical and epidemiological aspects.
In summary, the manuscript presents scientific results of considerable interest, but requires a series of methodological and structural improvements to increase its overall quality. My advice is to proceed with a thorough revision addressing these points before moving forward with publication, as, if properly revised, the manuscript could represent a significant contribution to the relevant scientific literature.
Author Response
Comment 1: First of all, I would like to express my sincere gratitude for the opportunity to contribute my feedback to the evaluation of your manuscript. I found the topic addressed to be extremely interesting and relevant to our field. The research presents several useful and promising insights that could lead to significant advancements in our sector. However, after a thorough reading, I believe there are some aspects, mainly methodological, that need to be improved and clarified in order to fully highlight the value of the proposed work. Below, I outline the main areas that could benefit from further exploration and revision.
Response 1: Thankyou for your detailed and very helpful review. This has aided us in refining our manuscript into a more clearly structured report. In particular, both the methodology and discussion sections have now been subdivided into sections to improve flow and readability.
Comment 2: Editing: There is no legend for the acronyms used in the summary tables, and the use of acronyms is inconsistent.
Response 2: All figure and table acronyms have either been removed or clearly defined
Comment 3: Title: I suggest making it clearer, possibly by shortening it but making it more specific in terms of the study conducted, e.g., retrospective observational study. Additionally, as in the rest of the text, I recommend removing "mellitus" from the definition of gestational diabetes, as it is now outdated in the scientific community. I advise the authors to review the titles of the most recent prestigious reference guidelines.
Response 3: Agreed. The title has been shortened and retrospective observational study type added. “Mellitus” has been removed and “GDM” changed to gestational diabetes throughout the manuscript
Comment 4: Abstract: There is a lack of a real interpretation in the conclusions regarding the clinical practice implications of the phenomenon studied, with potential suggestions and specific proposals. The results should include (given the type of study conducted) more quantitative data rather than qualitative descriptions. I suggest revising the text (lines 19 to 26) based on the previous comments.
Response 4: Agreed. A sentence has been added to the abstract conclusion regarding implications for sustainable health service planning (32-33). The sentence on future research is more specific (33-34). Relevant quantitative data has been added to the results (20-30).
Comment 5: Keywords: I recommend four or five keywords that are more focused on the main topics of the study; the current ones are too sensitive.
Response 5: Agreed. The following keywords have been removed from line 36: dietetics, telemedicine; COVID19; maternal health services
Comment 6: Objectives: I believe they should conclude the introduction (lines 79-88), while the last part (89-99) should be part of the methods section.
Response 6: Agreed. The last paragraph (previously 89-99) has been moved to the methods section (180-186).
Comment 7: Methods: This is the most controversial aspect and certainly requires more attention, namely the lack of a structured reporting method, such as the STROBE checklist, which could have been used to ensure the scientific validity and transparency of the study. Including this checklist as a supplementary file and citing it in the text would be crucial to improving the quality and reproducibility of the study, making it more transparent and replicable. In this regard (lines 100-175), following the checklist mentioned above, I suggest dividing Chapter 2 into subchapters to make it clearer and more fluid.
Response 7: Agreed. Chapter 2 is now divided into subchapters, taking care to incorporate the STROBE guidelines, with reference made to these (191-193). The STROBE checklist has also been included as a supplementary file at your suggestion
Comment 8: Results: Clear and well-reported. Discussion: From lines 320 to 350, it is unclear whether the citations at the bottom justify the entire section. I suggest expanding or at least clarifying the references in the text.
Response 8: Agreed. These references have been moved earlier within this section, expanded (342-349), and re-cited at the section end (375-377). Additional literature has also been added (357-361, 368-370).
Comment 9: The same applies, but even more so, from lines 351 to 388, which completely lack supporting references for comparison. This is the most lacking aspect in this section, which necessarily needs to be compared with other studies or different care settings.
Response 9: Agreed. Relevant literature using different care settings has been added to this section (381-383, 387-389, 404-406, 409-410, 417-421).
Comment 10: From lines 403 to 410, they could support the conclusions.
Response 10: Agreed. This section has been moved to the conclusions (457-467)
Comment 11: Limitations: Every research study has limitations that should be explored and discussed clearly and comprehensively. The manuscript does not have a section explicitly dedicated to limitations, and this warrants a dedicated section.
Response 11: Agreed. A sub-chapter has been added within the discussion (442-448).
Comment 12: References: Should be expanded in the critical points mentioned and updated for those older than twenty years, especially in terms of clinical and epidemiological aspects.
Response 12: Thankyou. We did not find any references to be older than twenty years. The oldest from 2004, is one of only two studies found to address the relevant point (58-60), and still relevant within current practice.
Reviewer 2 Report
Comments and Suggestions for Authors In the background summary the objective is written, however, this is not an objective but the activity that was carried out, I suggest writing the objective of the study indicating a verb appropriate to the type of study such as "determine or establish, etc.At the end of the introduction I suggest ending it with the research question, it is not enough to formulate the hypothesis, which of course is obvious.
It is important to mention the limitations of the study, since this gives guidelines for the consideration of other studies with similar characteristics.
I congratulate the authors for this idea of using an important data sample, which allows adequate support for the study. In addition to the way they have approached it
Author Response
Comment 1: In the background summary the objective is written, however, this is not an objective but the activity that was carried out, I suggest writing the objective of the study indicating a verb appropriate to the type of study such as "determine or establish, etc.
Response 1: Agreed. The objective has been changed to “determine” (12-13)
Comment 2: At the end of the introduction I suggest ending it with the research question, it is not enough to formulate the hypothesis, which of course is obvious.
Response 2: Agreed. The research question now ends the introduction (81-83), while the paragraph previously concluding the introduction has moved to the methods (180-186), per the suggestion from another reviewer.
Comment 3: It is important to mention the limitations of the study, since this gives guidelines for the consideration of other studies with similar characteristics.
Response 3: Agreed. A separate section has been added within the discussion (442-448).
Comment 4: I congratulate the authors for this idea of using an important data sample, which allows adequate support for the study. In addition to the way they have approached it
Response 4: Thankyou for your prompt and helpful review. We appreciate the time you have taken to examine this manuscript and your encouragement.
Reviewer 3 Report
Comments and Suggestions for Authors
The study introduces and evaluates a risk prioritized approach to providing medical nutrition therapy (MNT) for individuals with gestational diabetes mellitus (GDM) compared to a universal approach. This addresses the gap in determining the efficiency and effectiveness of resource allocation in clinical dietetics for GDM management. Based on the provided content of the article, here are the recommended improvements to the methodology and further controls the authors might consider:
1) Incorporating data on participants’ nutritional knowledge, dietary behaviors, and satisfaction with care would provide a more comprehensive evaluation of the effectiveness of the MNT service models.
2) Conduct further analysis to account for confounders, such as lifestyle changes (dietary patterns, physical activity) and differences in healthcare access, which may have influenced outcomes.
3) Evaluating long-term impacts of the RP versus universal MNT on maternal and child health outcomes post-partum would strengthen the conclusions about the sustainability of these models.
By addressing these aspects, the study could refine its methodology, control for potential biases, and offer more definitive conclusions regarding the efficacy and efficiency of risk prioritized MNT compared to universal MNT.
Author Response
Comment 1: The study introduces and evaluates a risk prioritized approach to providing medical nutrition therapy (MNT) for individuals with gestational diabetes mellitus (GDM) compared to a universal approach. This addresses the gap in determining the efficiency and effectiveness of resource allocation in clinical dietetics for GDM management.
Response 1: Thankyou for reviewing our manuscript. We have attempted to incorporate your “bigger picture” insight, which is highly relevant.
Comment 2: Based on the provided content of the article, here are the recommended improvements to the methodology and further controls the authors might consider: 1) Incorporating data on participants’ nutritional knowledge, dietary behaviors, and satisfaction with care would provide a more comprehensive evaluation of the effectiveness of the MNT service models.
Response 2: Agreed. This is mentioned in the limitations (440-441) and has been added to recommendations for future research in the study conclusions (468-469)
Comment 3: 2) Conduct further analysis to account for confounders, such as lifestyle changes (dietary patterns, physical activity) and differences in healthcare access, which may have influenced outcomes.
Response 3: Agreed. This has been added to the recommendations for future research in the study conclusions (470-471)
Comment 4: 3) Evaluating long-term impacts of the RP versus universal MNT on maternal and child health outcomes post-partum would strengthen the conclusions about the sustainability of these models.
Response 4: Agreed. This is now mentioned in the abstract (34-35), limitations (444-448) and recommendations for future research in the study conclusions (472-474)